# Advances in Research on Pig Salivary Analytes: A Window to Reveal Pig Health and Physiological Status

**DOI:** 10.3390/ani14030374

**Published:** 2024-01-24

**Authors:** Lixiang Zheng, Lidan Shi, Xiangzhe Wu, Panyang Hu, Ben Zhang, Xuelei Han, Kejun Wang, Xiuling Li, Feng Yang, Yining Wang, Xinjian Li, Ruimin Qiao

**Affiliations:** College of Animal Science and Technology, Henan Agricultural University, Zhengzhou 450046, China; zhenglx0712@163.com (L.Z.); sld1014@163.com (L.S.);

**Keywords:** salivary analytes, pig health, disease diagnosis, biomarker

## Abstract

**Simple Summary:**

The analysis of pig saliva, which is easily available and rich in bioactive ingredients, has been widely studied, showing potential in disease diagnosis and treatment for pig production management. Studies have shown that porcine saliva analytes have important application potential in disease detection and can be used as disease biomarkers and therapeutic monitoring indicators, as well as for prognostic evaluation. Through the changes in specific proteins, metabolites, and microbial composition in porcine saliva, porcine saliva analytes can provide a new strategy for early diagnosis, prognosis assessment, and treatment of diseases. This review introduces the composition and function of analytes in porcine saliva. Different analytes in porcine saliva are widely employed in diseases. As an indicator of pig health, saliva can be used as a noninvasive sample for disease detection and monitoring.

**Abstract:**

Saliva is an important exocrine fluid that is easy to collect and is a complex mixture of proteins and other molecules from multiple sources from which considerable biological information can be mined. Pig saliva, as an easily available biological liquid rich in bioactive ingredients, is rich in nucleic acid analytes, such as eggs, enzymes, amino acids, sugars, etc. The expression levels of these components in different diseases have received extensive attention, and the analysis of specific proteins, metabolites, and biological compositions in pig saliva has become a new direction for disease diagnosis and treatment. The study of the changes in analytes in pig saliva can provide a new strategy for early diagnosis, prognosis assessment, and treatment of diseases. In this paper, the detection methods and research progress of porcine salivary analytes are reviewed, the application and research progress of porcine salivary analytes in diseases are discussed, and the future application prospect is presented.

## 1. Introduction

In recent years, saliva has gained interest as a biological sample that can be used to analyze disease research [1]. Saliva is secreted from three major paired glands (parotid, submandibular, and sublingual glands) and from hundreds of minor salivary glands spread over most parts of the oral mucosa [2]. Saliva moistens the oral cavity; covers every structure inside, such as mucosa and teeth; and interacts with food constituents. Thereby, it plays a major role in the protection and maintenance of mucosa integrity in the upper alimentary tract and is required for mastication, the mineralization of teeth, and the control of microorganisms, as well as for taste perception and digestion [3]. The main component of saliva is water, comprising more than 99% [4]. Although saliva is predominantly a watery fluid, it also consists of a complex mixture of proteins, ions, and other organic compounds produced mostly by the salivary glands (Figure 1), with a small portion originating from the blood. Its collection is easy and noninvasive, does not require the use of specialized material, and can be performed by nontrained staff [5]. This fact is relevant especially in pigs, since the need for restraint in this species makes blood collection painful and stressful [6]. Studies have shown that about 20% of proteins in human saliva exist in plasma, and saliva can reflect the concentration of hormones, drugs, antibodies, viruses, and other components in serum or mouth to a certain extent, indicating the importance and potential of saliva as a biological liquid [7]. Ortin-Bustillo et al. have shown that the levels of certain proteins such as S100A8/A9 and/or A12 are significantly increased in pigs with diarrhea infected by *Escherichia coli*. These proteins are typically associated with various inflammatory conditions, immune-mediated diseases, and sepsis, and are considered as potential and interesting biomarkers [8]. The activity levels of adenosine deaminase (ADA) are significantly elevated in the saliva of animals with localized inflammation, gastrointestinal diseases, and respiratory system diseases [9]. Infection with *Streptococcus suis* leads to changes in salivary biomarkers related to stress, inflammation, oxidative redox status, and muscle damage in pigs, which can be considered potential biomarkers for this disease [10]. Therefore, this study summarizes the collection methods of salivary analytes, their roles in pig production, and their applications in disease diagnosis.

## 2. Methods of Saliva Collection and Detection

In pigs, saliva has many advantages as a biological sample. Saliva can not only provide analytes with different pieces of information, but the sample collection is also simple [11]. It can be obtained through noninvasive and usually easy procedures, and the sampling procedure does not cause pain. In addition, repeated specimens can be obtained anytime, anywhere, and without the need for specialized staff. Therefore, it is widely used in both the veterinary and human medical fields [12,13]. Secondly, in saliva collection, animals are in control, mainly because they have more flexible chewing and tongue movement and can better control the speed and amount of saliva outflow, and therefore saliva sampling can be concentrated in the mouth. In contrast to passive blood collection, saliva accurately reflects the state of the animal’s body and can be carried out in compliance with animal welfare regulations [14]. Moreover, blood samples are subject to contamination in young piglets, when changes in their tooth growth lead to infections. As a result, saliva can play an important role in animal disease detection instead of blood. In recent years, noninvasive body fluids such as animal saliva, urine, vaginal secretions, and feces have been proposed as alternatives to blood samples as testing sources for sex hormones and have shown more reasonable results in many mammals, such as pigs, sheep, buffalo, and so on [15]. In pigs, saliva has traditionally been used for stress assessment via the measurement of cortisol. When pigs are exposed to exogenous stress, the level of cortisol in the blood increases by 15% [16]. The correlation between salivary and plasma cortisol was stronger when long-term stable cortisol secretion was induced than when acute cortisol elevation was induced [17].

Different methods of saliva collection have different effects on pigs [18]. The selection of the collection method should also take into account the material, shape, size, and other factors of the collection instrument to ensure the quality of the collected saliva samples. One method involves hanging a cotton rope at shoulder height. To minimize sample contamination, the rope is hung away from fence walls, feeders, and waterers, or it is held by the operator. Oral fluids are transferred to a plastic bag attached to a 10 mL tube with an open-angle connector and squeezed out, extracting oral fluids from the rope. Another collection method involves using a sponge, which is held with forceps and exposed in the pig pen. After sample collection, the saliva is transferred into the saliva tube. The collection device is chewed on until visibly moist, allowing the animal to chew the sponge until it is fully saturated. The saturated sponge is then placed into a test tube and centrifuged at 3000 rpm for 10 min to obtain approximately 0.5–1 mL of saliva [19]. Rope sampling is usually used for weaned piglets, whereas sponge sampling is commonly used in farms for pigs of all ages except weaned piglets [20].

There are many advantages to saliva-based methods. Collected saliva can be directly frozen, and thus various enzymes in saliva (such as salivary amylase, and alkaline protease) as well as antimicrobial substances can be preserved in samples for a longer time. Simple saliva-based methods do not cause any harm to the body and do not induce stress in pigs. Additionally, saliva contains predominantly secretory antibodies, which hold special significance in the interpretation of immune function.

## 3. Overview of Saliva

### 3.1. Function of Saliva

Saliva testing in pigs is an emerging method that can be used for health monitoring, disease diagnosis, and drug residue detection, among others [21]. The rich biological information present in pig saliva enables the use of saliva testing for post-immunization antibody detection, early disease diagnosis, treatment efficacy assessment, and other applications. Compared to serum testing methods, saliva testing has the advantage of easy collection [22]. Saliva contains secretory antibodies that are produced rapidly, exhibit high stability, and possess active antigen-capturing abilities, making it a promising approach. Various analytes can be found in pig saliva, such as touch proteins, cortisol, estradiol sulfate, progesterone, IgG, and IgA. Several reports have already been published on the use of pig oral fluids for pathogen and antibody monitoring. The reported diseases that can be monitored using oral fluids include porcine reproductive and respiratory syndrome (PRRS) [23], porcine circovirus type 2 (PCV2), and swine influenza.

Saliva components include many types of contents, and different components perform different functions. In the driving function, saliva can protect the mouth from harmful microorganisms and irritants. It not only lubricates the oral tissue but also helps in the proper functioning of various other functions, such as chewing and swallowing, as well as protecting teeth and oral tissue [24]. The presence of inorganic nitrates can be detected in the daily diet of humans, such as many vegetables, fruits, and drinking water. In human blood circulation, after human metabolism and other processes, about 25% of the nitrate can be actively absorbed by salivary glands. The salivary glands will secrete it into saliva after absorption, and after circulating in the blood, the nitrate concentration in the saliva is about 10 times that of the blood [25]. Studies have shown that nitrates have a potential preventative effect on diseases such as obesity, diabetes, and heart disease [26,27]. Sialin (*Slc17a5*) is a transmembrane protein that is highly expressed in the acinar cells of the salivary gland. The transport of nitrates is dependent on sialin. The transport of many substances in the cell such as glutamate and aspartic acid also depends on sialoprotein. In addition, sialin helps to maintain the physiological function of acinar cells and plays a very important role in the normal physiological function of acinar cells [28]. Recently, studies have shown that the interaction between sialin and nitrate is specific to the salivary gland, and the interaction between sialin and nitrate has not been found in other organs [29]. Therefore, the interaction between nitrate and sialic acid is likely to play an extremely important role in maintaining the homeostasis of the salivary glands. Moreover, saliva is an important tool for measuring an individual’s physiology and disease status [30]. Studies have shown that the amount of oxytocin in the saliva of sows varies on different days after delivery [31]. Therefore, the physiological state of pigs can be revealed by detecting the content of different analytes in saliva.

### 3.2. Biomarkers in Saliva

Pigs utilize chemical signals for reproduction and various other behaviors, and the success of reproduction often depends on the changes in steroid pheromones found in boar saliva. Steroids have been shown to stimulate estrus behavior in gilts, aiding in the detection of estrus and determining the optimal timing for artificial insemination [32]. Steroids can bind to carrier proteins, and research suggests that steroids are present in the submaxillary saliva of boars. These salivary steroids can bind to carrier proteins in the nasal mucus of female pigs, ultimately triggering cascades of activity at the olfactory and endocrine levels. In addition to steroid pheromones, saliva also contains specific proteins that are believed to act as carrier molecules for these steroid pheromones [33].

In the process of pig production, the accuracy of estrus detection is an important factor affecting the reproductive performance of sows. The accuracy of estrus detection can have an impact on the conception rate of sows [34]. As a result, it affects the production efficiency and income of pig farms. Currently, the estrus detection methods used in pig farms are relatively traditional. In order to improve the timeliness and accuracy of estrus detection, saliva-based detection can be used as a new approach in swine production. The salivary proteins related to estrus were screened using the quantitative detection of oral salivary proteins in sows. It provides guidance for the accurate identification of estrus in sows [35]. This method involves screening saliva proteins associated with estrus [36], providing important references for accurately identifying the estrus of sows. By employing saliva-based detection, farm managers can make more accurate assessments of the estrus status in sows and take appropriate breeding measures, thereby enhancing the conception rate of sows and the overall production efficiency and income of pig farms [35]. In the period before estrus, there is a time frame in which replacement sows undergo ovarian development and the secretion of reproductive hormones. During this period, external stimuli such as exposure to a boar can induce ovulation in sows. To identify this pre-estrus stage in sows on pig farms, studies have examined potential biomarkers in saliva that can serve as candidates for the pre-estrus indicators of reproductive activity. These candidate biomarkers found in saliva include butyrate and 2HOvalerate, 5.79 ppm [36]. The research and validation of biomarkers in saliva have played a significant role in identifying the physiological status of sows and determining the optimal timing of boar exposure. This has greatly contributed to improving sow fertility and reproductive performance, ultimately maximizing the benefits for pig farms.

## 4. The Use of Saliva for Disease Diagnosis

### 4.1. Application of Porcine Salivary Protein Analytes for Disease Diagnosis

Oral fluid testing presents a chance to efficiently gather population-level disease data, addressing the challenges associated with pathogen transmission and timely information. In the past few years, noninvasive sampling techniques based on collecting oral fluid specimens have gained significant attention in the field of swine herd health management. The detection of pathogen levels in oral fluid samples has been proven to be a valuable method for improving the efficiency and cost-effectiveness of pathogen monitoring. More than 2000 proteins and peptides related to oral cancer and systemic diseases have been discovered in saliva [37]. Prims et al. identified new porcine salivary proteins and mammalian salivary proteins from porcine saliva, and there were quantitative differences in salivary proteins secreted by different salivary glands. Their study contributes to the search for potential biomarkers that can help in the early detection of pathology [38]. The salivary proteome has been used to assess the changes in its content in systemic immune diseases resulting from inflammation [39]. It reveals the great potential of proteomics in biomarker identification. Meningitis is a disease of concern. The occurrence of meningitis can cause changes in pig saliva and serum proteome. Meningitis is caused by *Streptococcus suis*, which is a Gram-positive bacterium that is considered to be one of the most important swine pathogens. Common swine diseases caused by *Streptococcus suis* can lead to significant increases in farm mortality and morbidity. Studies have shown that the average post-weaning mortality rate is 14% [40]. *Streptococcus suis* causes meningitis and induces changes in pig saliva and serum proteome [41]. Saliva can be used as a biomarker for health and welfare assessment and disease detection. Saliva is receiving more and more attention in animal health research [42]. The acquisition of saliva is simple and does not require considerable human and material resources. The sampling process of saliva does not require complex equipment, so saliva is helpful for large-scale sampling [1]. In a previous study, the analysis showed a total of 21 salivary analytes when pigs had meningitis. To reduce the mortality rate associated with this disease, it is crucial to detect and treat SHF in its early stages. Saliva, as an alternative diagnostic medium, has gained widespread attention due to its ease of collection and noninvasive nature. In addition, saliva collection reduces the risk of infection, and studies have shown that saliva contains approximately 30% of the biomolecules present in the blood.

### 4.2. Application of Porcine Salivary Nucleic Acid Analytes for Disease Diagnosis

In early disease diagnosis, the analysis of nucleic acid components in pig saliva can lead to the detection of the presence and quantity of various pathogens such as viruses [43]. Salivary small extracellular vesicles can carry a variety of bioactive molecules and are rich in small noncoding RNA [44]. Recent studies have shown that miR-512-3p and miR-412-3p are upregulated in salivary small extracellular vesicles in patients with oral squamous cell carcinoma [45]. The expression level of the infected group was alleviated. NEAT1 is an inflammatory regulator that promotes the activation of inflammasome in macrophages [46]. NEAT1 is an important component of the antiviral immune response, and NEAT1 lncRNA is related to the immune system response. In the future, new targets may be discovered that can be used for different diagnostic measures or treatments based on disease severity. For example, in early infections of pathogens such as the porcine epidemic diarrhea virus and viral nucleic acids can be detected in pig saliva [47]. This has a potential value for early disease diagnosis. Additionally, the nucleic acid components in pig saliva can also be used in infectious disease monitoring [48]. Vaccine response assessment and production efficiency monitoring also play important roles. By collecting saliva samples from pig populations and detecting disease-related nucleic acids, the monitoring and tracking of disease spread can allow for the assessment of the extent of disease exacerbation or mitigation. This helps guide the development of corresponding prevention and control strategies [49]. The specific nucleic acid components in pig saliva can also be used to evaluate the effectiveness of vaccines in pig populations [50]. This helps to assess vaccine efficacy and improve vaccine design to enhance the immune status of pig populations. The nucleic acid components in pig saliva can be used to monitor the relationship between production performance and diseases. By analyzing specific genes or physiological markers in saliva samples, we gain insights into the health status, sexual maturity, and reproductive capacity of pigs.

### 4.3. Application of Metabolites in Pig Saliva to Disease

Metabolites in pig saliva primarily consist of small-molecule metabolites [51]. Salivary metabolites participate in various cellular functions, such as the direct regulation of gene expression. The origin of porcine susceptibility to stress is a mutation in the RYR1 gene [52]. High levels of stress increase susceptibility to disease, decrease life expectancy, impair growth and reproduction, cause body damage and abnormal behavior, and decrease meat quality [53]. Pigs respond to stress with increased adrenal cortical activity and corticosteroid levels [36]. Cortisol and corticosterone are the main glucocorticoids secreted in pig saliva [54]. Although both cortisol and corticosterone are easily detected in saliva, cortisol detection is more sensitive, and its results are less variable. Therefore, cortisol can be used as the main stress biomarker in pig saliva, while corticosterone can be used as a confirmation biomarker [55]. They serve as effector molecules of molecular events that contribute to disease development [56]. Saliva contains both endogenous and exogenous metabolites [57]. The concentration of metabolites in saliva is correlated with the plasma concentration. Steroids easily enter saliva from plasma [58], so testing for steroid levels can provide a basis for saliva diagnosis and provide a noninvasive method of monitoring. In addition [59], small molecules such as hormones can also be determined in saliva and used as indicators of human health and disease status [12]. Saliva has proven to be a valuable analytical tool in metabolomics. Recently, lipidomics has become a new subfield of metabolomics [60]. The analysis of lipidomics includes the structural and functional components of lipids, as well as their interactions with other lipids, proteins, and metabolic pathways [61]. The lipid set of the organ can provide biomarkers for metabolic changes that occur during disease and for early detection of the disease [62]. The main source of salivary metabolites is the oral metabolic pathway [63]. The detection and analysis of these metabolic products can aid physicians in diagnosing certain oral diseases, among other things. Additionally, in disease monitoring, the metabolites found in pig saliva can reflect the progression of a disease and the efficacy of treatments [64]. By regularly examining and analyzing the metabolic products in pig saliva, it is possible to monitor changes in diseases and evaluate the effectiveness of treatments. Prevention measures can also be guided by the development of strategies for preventing and treating diseases.

### 4.4. Application of Microorganisms in Pig Saliva to Disease

Saliva is sterile when secreted into the mouth [65]. However, a variety of microbiomes are present in a saliva sample [66]. Thus, the salivary microbiome has been shown to be a colony of bacteria shed from the surface of the mouth and is temporarily stable in individuals with oral health [67]. The composition of salivary microbiota in health is influenced to some extent by environmental factors [68]. Dental caries is the most common chronic infectious oral disease in children and adolescents worldwide. If it is not prevented early, it will lead to the destruction of hard tooth tissues. *Streptococcus mutans* and *Streptococcus Sothoni* are the strains most commonly associated with tooth decay in children. *Streptococcus mutants* are particularly cariogenic. In addition to mutant Streptococcus, *lactobacillus* also plays a role in the progression of dental caries. Therefore, mutant *streptococcus* and *Lactobacillus* can be used as markers of early childhood caries (ECC). In addition, the saliva microbiota is influenced by the stage of tooth development, and the early life development of the saliva microbiota is a coordinated process [69], which is affected by ecological perturbations such as delivery mode, breastfeeding duration, and antibiotic treatment. The human rhinovirus (HRV) is the primary cause of viral gastroenteritis worldwide [70]. The HRV is an enteric pathogen that can be transmitted through the fecal–oral route [71]. Research has shown that the HRV can replicate in salivary tissues [72]. Research has found a certain correlation between the microbial composition in pig saliva and oral diseases such as dental caries and periodontal disease [73]. By analyzing the microbial content in pig saliva, the development and treatment effectiveness of oral diseases can be monitored, allowing for the treatment of oral health diseases at an early stage. The microbial composition in pig saliva is also closely associated with respiratory diseases in pigs, such as swine influenza and porcine reproductive and respiratory syndromes [47]. Analyzing the microbial content in pig saliva can enable timely diagnosis and facilitate early treatment, which is crucial for improving productivity in the pig industry. By monitoring the microbial content in pig saliva, it is possible to track the development of respiratory infections and implement appropriate preventive and treatment measures [74] (Table 1).

## 5. Discussion

As a biological fluid, porcine saliva is rich in biologically active ingredients and has many physiological functions such as increasing immunity and antibacterial and anti-inflammatory functions, as well as promoting wound healing. Among nonblood-derived body fluids, saliva has the advantages of being noninvasive and cost-effective, and it also contains a variety of proteins, DNA, RNA, various metabolites, and microbial communities, which can be used as molecular biomarkers for the early detection of disease, disease monitoring, and personalized and precise treatment [81,82]. In recent years, an increasing number of studies have discovered the potential of pig saliva for disease application [83]. Here, we described the application and progress of saliva sampling in pigs for disease detection and treatment. By analyzing protein, nucleic acid, metabolites, and microbial composition in pig saliva, one can carry out the early diagnosis, prediction, and monitoring of diseases. Kobayashi et al. have shown that pig saliva inhibits the replication of influenza virus [84] and has certain preventive and therapeutic effects. Pig saliva also contains a variety of enzymes and immunoglobulins [85], which can inhibit the growth of bacteria in the mouth and effectively prevent the occurrence of oral diseases. Certain components such as immunoglobulin and antimicrobial peptides in pig saliva are known to enhance intestinal immunity and promote intestinal health.

Pigs are important livestock animals, and their health is crucial to the sustainable development of the pig industry. In recent years, saliva has become an effective tool for the detection of the physiological and pathological statuses of humans and animals in addition to plasma, serum, and urine [37]. The early diagnosis and timely monitoring of diseases are of great significance to effectively control the spread of diseases, improve production efficiency, and ensure the healthy development of animals in the livestock and poultry industry. At present, the collection of saliva samples in animals is mainly used for swine disease detection, including the antigen and antibody detection of swine disease pathogens such as the porcine reproductive and respiratory syndrome virus and the swine influenza virus [86]. The rich biological information in pig saliva makes saliva detection useful for antibody detection after immunization, the early diagnosis of disease, and the assessment of treatment efficacy [87]. Compared with the serum detection method, in addition to the advantages of convenient collection, the secreted antibodies in saliva have a fast production speed, high stability, and active capturing ability of antigens, and therefore they have a good application prospect.

Pig saliva nucleic acid analytes have great potential for disease application but still face some challenges. For example, the standardization of techniques and the sensitivity and specificity of detection methods still need to be further improved. Studies are needed to investigate the association between salivary analytes and different diseases, as well as emerging technologies such as CRISPR-Cas9 [88], to further improve the application of porcine salivary analytes in disease prevention and surveillance.

## 6. Conclusions

In this review, we described the composition of porcine saliva. As a biological fluid, porcine saliva is rich in biologically active components. Different components of pig saliva travel different functions. The expression levels of proteins, nucleic acids, and microorganisms in salivary analytes in disease have received considerable attention. The analysis of specific proteins, metabolites, and microbial composition of porcine saliva has become a new direction for disease diagnosis and treatment. In the future, with the continuous progress of science and technology, the sensitivity and specificity of pig saliva analysis will be further improved. The adoption of porcine saliva analytes is anticipated to lend credence to disease detections in pigs.

## Figures and Tables

**Figure 1 animals-14-00374-f001:**
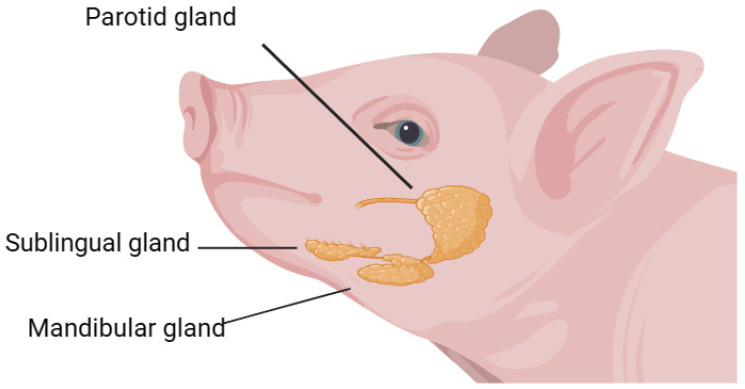
Diagram of a pig’s salivary glands.

**Table 1 animals-14-00374-t001:** Composition and function of analytes in saliva.

Analytes	Types	Function	References
protein	lactoferrin	immune modulator	[75]
	CRP, CK-MB, sCD40	metabolism; immune	[26]
	histatin-1 (HTN1)	antifungal	[76]
nucleic acid	SAT(mRNA)	convey information	[77]
	IL-8(mRNA)		
metabolites	peptides	metabolism; regulation of physiological balance	[78,79]
	vitamins		
	organic acids		
	thiols		
microorganisms	*N. elongata*	cancer biomarker	[80]
	*S. mitis*		

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
