# Peer review of "Advances in Research on Pig Salivary Analytes: A Window to Reveal Pig Health and Physiological Status"

_animals, 2024, doi:10.3390/ani14030374_

Round 1

Reviewer 1 Report

Comments and Suggestions for Authors

1.Abstract, authors are suggested to start broad in the general background, and then increase the necessity for research. This contents should be improved.

2. Additional reports regarding the use of salivary analytes in diseases could be included in the introduction.

3.The conclusions need to be more concise. The authors are suggested to highlight the important content of the article and make a prospect for future research.

4. There is at least one grammatical error in the manuscript, for example, in line 44, the name "Bandhakavi" should not be placed at the beginning of the sentence. Please check the manuscript carefully and improve the language.

5. Some sections, such as lines 100 and 143, should include references.

Comments on the Quality of English Language

Please check the manuscript carefully and improve the language.

Author Response

Dear editor,

  Thank you for your comments on the revision of the manuscript. Based on your revised comments, We have checked the manuscript and revised it. The modified content is shown as follows:

  1. In page 1, Abstract, authors are suggested to start broad in the general background, and then increase the necessity for research. Maybe this part can be improved.

Response 1: Thanks for your suggestion. We have modified the abstract to add the background of saliva as an important exocrine fluid to justify the need for porcine saliva research. Now lined of 19-21.

  1. Additional reports regarding the use of salivary analytes in diseases could be included in the introduction.

Response 2: Thanks for your suggestion. At the introduction of the review, in line 50-61 of the article, we added salivary analytes to the study of disease.

  1. The conclusions need to be more concise. The authors are suggested to highlight important the important content of the article and make a prospect for future research.

Response 3: Thanks for your suggestion. At the conclusion of the review, we have modified some of the content to make the conclusion more concise. The specific content has been presented in the article. Lined of 335-339.

  1. There is at least one grammatical error in the manuscript, for example, in line 44, the name "Bandhakavi" should not be placed at the beginning of the sentence. Please examine the manuscript carefully. Pay special attention to the need for professional revision of the language.

Response 4: Thanks for your suggestion. We have modified the name "Bandhakavi" at the beginning of the sentence, and now correct sentence lined of 46.

  1. Some sections, such as lines 100 and 143, should include references.

Response 5: Thanks for your suggestion. We added references about some sections, and now lined of 112 and 156.

Reviewer 2 Report

Comments and Suggestions for Authors

Dear authors,

I have written your manuscript entitled "Advances in research on pig salivary analyses: A window to reveal pig health and physiological status" with great interest since I have been working in the field on porcine salivary proteomics for 10 years now.

The review remains superficial and would certainly benefit from the inclusion of in-depth analyses of the available literature. For example, when describing the collection method of saliva in the pig, it is important to mention that blood contamination is possible in young piglets, which is due to the changing of teeth. As a result, blood analytes could interfere with the results. 

Another example: Stress has briefly been dealt with, but this topic could be elaborated. Stress is very important in pigs. How can we determine the level of stress? What about the difference between blood and saliva? What about acute and chronic stress?

The review is mainly focussed on pig health and disease, but only to a lesser extent to physiological status. This parameter together with wellbeing is very important. They determine the economical outcome of pig farming. So please, elaborate on this topic. 

Proteomics in another important tool to analyse porcine saliva. Please include the recent advances in this field. In humans, over 3000 proteins have been identified in saliva. In contrast, only around 1 percent of these have been found in porcine saliva. This probably means that a lot of progress can be made here.

Speaking of human saliva. Please try to limit the references to human saliva as much as possible, and focus as much as possible on the pig.

Minor comments:

- Reference 24 is missing in the reference list. Please check whether the references in the list are still matching the numbers in the text.

- Is it correct that a ":" should be placed in between "analytes" and  "a window" in the title, as I have suggested above.

- Figure 1 has been adapted from the atlas of Popesko. Is it allowed to reproduce this figure without permission? At least reference should be made.

Author Response

Revision 2

Dear Reviewer,

Thank you for your comments on the revision of the manuscript. Based on your revised comments, We have checked the manuscript and revised it. The modified content is shown as follows.

The review remains superficial and would certainly benefit from the inclusion of in-depth analyses of the available literature. For example, when describing the collection method of saliva in the pig, it is important to mention that blood contamination is possible in young piglets, which is due to the changing of teeth. As a result, blood analytes could interfere with the results.

Response 1: Thanks for your suggestion. In line 72, we added blood contamination is possible in young piglets. Then put forward saliva can play an important role in disease detection instead of blood.

Another example: Stress has briefly been dealt with, but this topic could be elaborated. Stress is very important in pigs. How can we determine the level of stress? What about the difference between blood and saliva? What about acute and chronic stress?

Response 2: Thanks for your suggestion. In line78-81, we added the effects of stress levels on cortisol content, the difference between blood and saliva and acute stress and chronic stress on increased cortisol level of stress.

The review is mainly focussed on pig health and disease, but only to a lesser extent to physiological status. This parameter together with wellbeing is very important. They determine the economical outcome of pig farming. So please, elaborate on this topic.

Response 3: Thanks for your suggestion. In line 136-139, we added the content of sow saliva to reflect the physiological state of pig.

Proteomics in another important tool to analyse porcine saliva. Please include the recent advances in this field. In humans, over 3000 proteins have been identified in saliva. In contrast, only around 1percent of these have been found in porcine saliva. This probably means that a lot of progress can be made here.

Response 3: Thanks for your suggestion. In chapter 4.1, we added recent advances in proteomics in salivary studies to detect disease.

Speaking of human saliva. Please try to limit the references to human saliva as much as possible, and focus as much as possible on the pig.

Response 3: Thanks for your suggestion. We reduced the reference to human saliva. For example, In line 116, we deleted the reference to human saliva.

Minor comments:

- Reference 24 is missing in the reference list. Please check whether the references in the list are still matching the numbers in the text.

- Is it correct that a ":" should be placed in between "analytes" and "a window" in the title, as I have suggested above.

- Figure 1 has been adapted from the atlas of Popesko. Is it allowed to reproduce this figure without permission? At least reference should be made.

Response 6: Thanks for your suggestion. We added the missing reference and ":" placed in between "analytes" and "a window" in the title. More, Figure 1 is a self-created picture and I've already explained that.

Reviewer 3 Report

Comments and Suggestions for Authors

The topic is undoubtedly interesting, but is not presented in sufficient detail in the manuscript. Unfortunately, the authors only briefly discuss individual facts about the usefulness of saliva. Instead of claims about pigs, human facts are often cited (e.g. oral cancer, periodontitis patients, ...). The article is poorly structured and the text contains many typos. The manuscript lacks facts about the inadequacy of saliva as a material for laboratory diagnostics.

-line 58: that saliva is stable at room temperature? The statement is incorrect because when saliva is used for molecular diagnostics, it must be administered as soon as possible after collection and brought to the laboratory on ice for further processing.

- chapter 2: the methods of saliva collection and detection are not well described; the method of sampling according to pig category and the material used for sampling is missing.

- line 73: insufficiently explained; it is necessary to specify the sampling with rope.

-line 103: why humans? Find a reference for pigs. All citations referring to humans should be removed from the manuscript. The focus should be on pigs.

- line 102: The function of saliva does not fall under the chapter application of saliva analyte evidence.

- chapter 3.2: No specific methods and diagnostic kits are mentioned in the chapter. This is missing in a review article.

- chapter 4: A table listing which diseases can be diagnosed with which detection method is missing

- line 194-199: What does this have to do with pigs?

- line 217: Where is the connection with the use of saliva for diseases?

- line 254-266: What does this have to do with pigs?

- line 292: Is there oral cancer in pigs?

Comments on the Quality of English Language

Typos should be corrected.

Author Response

Revision 3

Dear Reviewer,

Thank you for your comments on the revision of the manuscript. Based on your revised comments, We have checked the manuscript and revised it. The modified content is shown as follows.

-line 58: that saliva is stable at room temperature? The statement is incorrect because when saliva is used for molecular diagnostics, it must be administered as soon as possible after collection and brought to the laboratory on ice for further processing.

Response 1: Thanks for your suggestion. I already know from study that saliva is unstable at room temperature. According to your suggestion, I have deleted the wrong statement in manuscript.

- chapter 2: the methods of saliva collection and detection are not well described; the method of sampling according to pig category and the material used for sampling is missing.

Response 2: Thanks for your suggestion. In line 93-95, we added the method of sampling according to pig category and the material used for sampling.

- line 73: insufficiently explained; it is necessary to specify the sampling with rope.

Response 3: Thanks for your suggestion. In line 88-90, we specify the sampling with rope and we have improved the content of chapter 2 to make the content of this part more applicable to the whole article.

-line 103: why humans? Find a reference for pigs. All citations referring to humans should be removed from the manuscript. The focus should be on pigs.

Response 4: Thanks for your suggestion. We have removed the reference to human saliva and added the functional application of the porcine saliva analyte.

- line 102: The function of saliva does not fall under the chapter application of saliva analyte evidence.

Response 5: Thanks for your suggestion. I have changed “Application of saliva analyte detection” to “Overview of saliva” of chapter 3.

- chapter 3.2: No specific methods and diagnostic kits are mentioned in the chapter. This is missing in a review article.

Response 6: Thanks for your suggestion. In line 155-157, we added specific methods for detecting estrus in sows.

- chapter 4: A table listing which diseases can be diagnosed with which detection method is missing.

Response 7: Thanks for your suggestion. The tables in this review mainly show the classification of salivary analytes and the presentation of their representative substances and functions. According to your suggestions, we have made supplements in chapter 4.  The application of the components of saliva analytes to different diseases and the methods of disease detection were supplemented.

- line 194-199: What does this have to do with pigs?

- line 217: Where is the connection with the use of saliva for diseases?

- line 254-266: What does this have to do with pigs?

- line 292: Is there oral cancer in pigs?

Response 8: Thanks for your suggestion. We have deleted the parts that are not relevant to the research content of this film, and improved the link between pig saliva and disease.

Comments on the Quality of English Language: Typos should be corrected.

Response 9: Thanks for your suggestion. For the English writing of the whole review, we have improved our English by seeking professional help.

Reviewer 4 Report

Comments and Suggestions for Authors

The paper is interesting and deals with a topic of current importance. There some some changes that can improve the manuscript such as:

-Point 2. 

*Line 52 you can put together the two sentences: "providing diverse information and the use of saliva.."

*Line 59 delete "it can stable at room temperature for several weeks", since many components are not stable so long time at room temperature

*Lines 68 and 69 delete "before collecting saliva, it can be made in a state of hunger to stimulate the secretion of saliva", since this is not recommended

*Line 73 instead "The first method" use "One method" and lin line 76 insteand "The second" use Another method

*Lines 78 and 79 delete "the aid of a funnel" and delete reference 11 since it is not from pigs, you can change by reference 4 for example.

*Line 88: delete reference 13 since it is not specific from pigs and use for example reference 7

*Lines 89-99 can be moved to the point 3.2.

-Point 3.

*3.2. Put "Biomarkers in saliva" as title

*lines 131 to 136 can be deleted

-Point 4.

*Lines 171-178 can be at the begining of the point 4.1. and then please put clearly the different diseases that can be detected in saliva and put together points 4.1 and 4.2. 

Also please deleted 163-166 since AD is a disease from humans and also 178-182 since there are references and applications of humans.

*try to put 4.3. in  a more clear way

-Point 5. 

Delete 292-294 since oral cancer is a big problem in humans but non in pigs

-Point 6.

"In this review, we provide information about the composition and applications of porcine saliva.

Author Response

Revision 4

Dear Reviewer,

Thank you for your comments on the revision of the manuscript. Based on your revised comments, We have checked the manuscript and revised it. The modified content is shown as follows:

-Point 2.

*Line 52 you can put together the two sentences: "providing diverse information and the use of saliva."

*Line 59 delete "it can stable at room temperature for several weeks", since many components are not stable so long time at room temperature

*Lines 68 and 69 delete "before collecting saliva, it can be made in a state of hunger to stimulate the secretion of saliva", since this is not recommended

*Line 73 instead "The first method" use "One method" and lin line 76 insteand "The second" use Another method

*Lines 78 and 79 delete "the aid of a funnel" and delete reference 11 since it is not from pigs, you can change by reference 4 for example.

*Line 88: delete reference 13 since it is not specific from pigs and use for example reference 7

*Lines 89-99 can be moved to the point 3.2.

Response 1: Thanks for your suggestion. We have deleted the references to unrelated references, improved and changed the content you mentioned. Moreover, we have reviewed and refined the scientific content cited in the paper. We re-examined the structural problem and moved lines 89-99 to the point 3.2 as you suggested.

-Point 3.

*3.2. Put "Biomarkers in saliva" as title

*lines 131 to 136 can be deleted

Response 2: Thanks for your suggestion. We have put "Biomarkers in saliva" as title of 3.2 and deleted the content of lines 131 to 136.

-Point 4.

*Lines 171-178 can be at the begining of the point 4.1. and then please put clearly the different diseases that can be detected in saliva and put together points 4.1 and 4.2.

Also please deleted 163-166 since AD is a disease from humans and also 178-182 since there are references and applications of humans.

*try to put 4.3. in a more clear way

Response 3: Thanks for your suggestion. We have deleted the content you mentioned and listed the diseases that porcine salivary proteomics can detect. For example, meningitis in pigs can cause changes in porcine salivary proteomics. Moreover, we have added and refined the content of 4.3 and the structural issues you mentioned to make this review more complete.

-Point 5.

Delete 292-294 since oral cancer is a big problem in humans but non in pigs

Response 4: Thanks for your suggestion. We have deleted line 292-294.

-Point 6.

"In this review, we provide information about the composition and applications of porcine saliva.

Response 4: Thanks for your suggestion. We modified the composition and application of porcine saliva in this review, supplemented and improved the content, and re-optimized the structure and organization of the paper. We sincerely thank you for your guidance and look forward to improving the paper through these improvements. Thank you again for your time and feedback.

Reviewer 5 Report

Comments and Suggestions for Authors

The manuscript is poor in both scientific knowledge and writing. A review should include the most actual knowledge in the field with a historical background. 

There are plenty of statements not supported by scientific knowledge in the whole manuscript.

Moreover information shown in the different sections is mixed and not focussed to the main idea of each section.

Since the state of the art is not fully addressed, the scientific contribution is vague.  

Comments on the Quality of English Language

The english language is not the limitation of the paper but the scientific quality. However, the manuscript will be improved if a native english speaker review the paper.

Author Response

Revision 5

Dear Reviewer,

Thank you very much for taking the time to review our paper and for providing valuable feedback. We highly appreciate the issues you've highlighted and are committed to making substantial improvements to meet the high standards of quality.

The manuscript is poor in both scientific knowledge and writing. A review should include the most actual knowledge in the field with a historical background.

There are plenty of statements not supported by scientific knowledge in the whole manuscript.

Moreover information shown in the different sections is mixed and not focussed to the main idea of each section.

Since the state of the art is not fully addressed, the scientific contribution is vague.

Comments on the Quality of English Language:

The english language is not the limitation of the paper but the scientific quality. However, the manuscript will be improved if a native english speaker review the paper.

Regarding the concerns about poor English expression and insufficient scientific knowledge, we will take the following steps to enhance our paper:

1.Language Expression: We have sought assistance from professional editors or peers to refine the language, ensuring accuracy and clarity in terms of grammar, spelling, and sentence structure.

2.Scientific Knowledge: We have examined and refined the scientific content referenced in the paper, ensuring all statements are well-supported and properly cited to enhance the credibility and scientific validity.

3.Logical Structure: We have re-evaluated the paper's structure and organization, ensuring each section tightly aligns with the main theme, presents ideas clearly, and avoids mixed information.

We genuinely appreciate your guidance and look forward to refining the paper through these improvements. Thank you once again for your valuable time and feedback.

Sincerely,

Lixiang Zheng

Round 2

Reviewer 2 Report

Comments and Suggestions for Authors

Dear authors,

Thank you for revising your manuscript. It has certainly improved.

I am, however, surprised that the manuscripts by Prims S et al. on the porcine proteome are not mentioned.

Comments on the Quality of English Language

Please have a critical look at the text and specifically the grammar/built of the sentences. For example, As a biological liquid, pig saliva contains rich biological active ingredients, and has many physiological functions such as high immunity, antibacterial and anti-inflammatory, and promoting wound healing. This sentence should read as follows: As a biological fluid, porcine saliva is rich in biologically active ingredients, and has many physiological functions such as increasing the immunity, antibacterial and anti-inflammatory functions, and promoting wound healing.

Another exampleSecondly, in saliva collection, the animal is in a dominant position, and compared with passive blood collection, it truly reflect the state of the animal body, and can be carried out in compliance with regulation requirements such as animal welfare. What is meant with "dominant position"? "it truly reflects": The "it" refers to saliva, but not saliva collection. 

Author Response

Dear reviewers,

Thank you very much for your thorough review and valuable feedback on our manuscript titled "Advances in research on pig salivary analytes: a window to reveal pig health and physiological status." We appreciate the time and effort you have invested in providing constructive comments to enhance the quality of our work. We have carefully considered each of your suggestions and made corresponding revisions to address the issues raised. Below is a summary of our responses to each of the points raised:

Response 1: Thanks for your suggestion. In line 184-187, we have added to the recent work of Prims S et al in porcine salivary proteomics. Prims et al. identified new porcine salivary proteins and mammalian salivary proteins from porcine saliva, and there were quantitative differences in salivary proteins secreted by different salivary glands. The study could help in the search for potential biomarkers that could help in the early detection of pathology

Response 2: Thanks for your suggestion. In line 69-74, we have improved the level of English language.

Response 3: Thanks for your suggestion. In line292-294, we have improved the level of English language.

Reviewer 3 Report

Comments and Suggestions for Authors

The authors took into account all my comments.

Author Response

Dear Reviewer,

Thank you for your comments and support.

Best regards.

Lixiang Zheng